# Spatial Position and Anatomical Characteristics Associated with Impacted Third Molars Using a Map-Reading Strategy on Cone-Beam Computed Tomography Scans: A Retrospective Analysis

**DOI:** 10.3390/diagnostics14030260

**Published:** 2024-01-25

**Authors:** Djalma Maciel de Lima, Cyntia Rodrigues de Araújo Estrela, Cristiane Martins Rodrigues Bernardes, Lucas Rodrigues de Araújo Estrela, Mike Reis Bueno, Carlos Estrela

**Affiliations:** 1Department of Oral Biology, School of Dentistry, Evangelical University of Goiás, Anápolis 75083-515, Brazil; djalmamaciel@terra.com.br (D.M.d.L.); cristiane.bernardes@unievangelica.edu.br (C.M.R.B.); estrelalucas4@gmail.com (L.R.d.A.E.); 2Center for Radiology and Orofacial Images, Diagnostic Imaging Center, Cuiabá 78043-272, Brazil; mikebueno@terra.com.br; 3Department of Endodontics, School of Dentistry, Federal University of Goiás, Goiânia 74605-020, Brazil; estrela3@terra.com.br

**Keywords:** anatomy, impacted third molar, mandibular canal, cone-beam computed tomography, surgery, inferior third molar

## Abstract

(1) Background: This study assessed the spatial position and anatomical features associated with impacted third molars through a map-reading strategy employing cone-beam computed tomography (CBCT). (2) Methods: The positioning of impacted third molars on CBCT was assessed using Winter’s and Pell and Gregory’s classifications. External root resorption in mandibular second molars was categorized according to Herman’s classification. Additionally, the relationship between the mandibular third molar root apex and the mandibular canal was examined. Comparative statistical analysis was conducted using Fisher’s exact test, with a significance level considered as 5%. (3) Results: The results indicated that, based on Winter’s classification, 48.06 % of impacted teeth were positioned mesioangularly. Employing Pell and Gregory’s classification, 43.22% of the impacted molars fell into positions B and C, with 54.2% classified as Class II. A notable 69.7% of teeth exhibited no contact between the root apex and the mandibular canal, and external root resorption in the distal aspect of the second molar was absent in 88.7% of cases. (4) Conclusions: Utilizing the map-reading strategy with CBCT scans to assess the anatomical positions and characteristics of impacted third molars enhances professional confidence and sets a standard for quality and safety in the surgical procedure for patients.

## 1. Introduction

The included teeth are those positioned within the bone/mucosal tissues during their physiological eruption. Impaction primarily arises from factors such as inadequate dental arch length, incorrect tooth angulation, delayed facial growth, delayed mineralization of the third molar, as well as the presence of pathological lesions that mechanically impede the eruption path. Additionally, the impaction can be influenced by the presence of excessively large teeth and dense bone/mucous tissue [1,2].

Impacted teeth can induce changes in adjacent structures, giving rise to the development of pericoronitis, primary and/or secondary dentition crowding, tumors, odontogenic cysts, associated periodontal defects, caries, as well as myofascial and neurogenic pain [3]. The extraction of mandibular third molars is a well-executed dental office procedure, necessitating meticulous planning to minimize the occurrence of severe pain, edema, discomfort, and/or dysfunction, whether temporary or permanent, the latter resulting from potential injury to the inferior alveolar nerve [4,5,6].

Prior to any intervention, the planning of third molar extraction involves a thorough case study. A comprehensive examination of the clinical presentation, including visual physical examination and palpation, in conjunction with imaging tests, facilitates the diagnosis and assessment of the relationship between these teeth and adjacent structures, such as the mandibular canal and lower second molars. This preemptive approach aims to prevent potential accidents during the intraoperative and postoperative periods, thereby mitigating complications [4].

Among the available imaging options, panoramic radiography has been recommended as the preferred standard method for investigating third molars, as it offers a comprehensive view of both the teeth and bone structures [7,8,9]. The study by Kim et al. [10] conducted an evaluation of the preoperative position of wisdom teeth using panoramic radiography. They observed that when the root of the impacted inferior third molar is embedded into the lingual cortical plate, a periapical band-like radiolucent sign may manifest in the panoramic image. According to the authors, this radiolucent sign can be valuable for predicting the root position and assessing surgical risks. However, studies have underscored the limitations of panoramic radiography, as it produces a two-dimensional image of three-dimensional structures, leading to the overlap of underlying anatomical structures. In contrast, cone-beam computed tomography (CBCT) has been identified as a superior alternative, providing more accurate information with minimal distortion when compared to conventional imaging exams [11,12,13,14,15,16,17,18].

The position of mandibular third molars is meticulously examined before surgical procedures to mitigate the serious risks of injuring the inferior alveolar nerve. In this context, the appropriate application of advanced imaging resources, including cone-beam computed tomography with navigation dynamics and sophisticated software, can prevent undesirable surgical complications. Therefore, this study assessed the spatial position and anatomical characteristics associated with impacted mandibular third molars using a map-reading strategy based on cone-beam computed tomography (CBCT) scans.

## 2. Materials and Methods

The present study utilized a sample of 200 cone-beam computed tomography scans from patients of both genders, ranging in age from 18 to 80 years. The scans were selected from a private clinic database, specifically the Center for Radiology and Orofacial Images of Cuiabá, Brazil (CROIF). These patients were referred to the radiology service for various diagnostic purposes between January 2015 and December 2020. This research was part of a more extensive study, which received approval from the Research Ethics Committee of the Federal University of Goiás (CAAE: 06486919.0.0000.5083).

The inclusion criteria for imaging examinations involved cases displaying unerupted or impacted mandibular third molars alongside the presence of mandibular second molars. Exclusion criteria comprised imaging tests featuring images of orthodontic appliances, teeth with internal inflammatory resorption, bone changes associated with systemic diseases, as well as benign and/or malignant neoplasms in the mandible.

CBCT images were acquired using the CS 8100 3D CBCT system (Carestream, Marne-la-Valée, France). The system was calibrated to a thickness of 0.15 mm, a field of view (FOV) of 8 × 9 cm, and a voxel size of 75. Tube voltage was set at 90 kVp, and the tube current was 3 mA, with an exposure time of 15 s. The acquired tomographic data were converted into DICOM (Digital Imaging and Communications in Medicine) format. Subsequently, the images were examined using e-Vol DX software 6.0 (CDT Software; São José dos Campos, SP, Brazil) [16]. The was conducted on a PC workstation equipped with an Intel i7-7700K processor 4, 20 GHz (Intel Corp., Santa Clara, CA, USA), an NVIDIA GeForce GTX 1070 Graphics Card (NVIDIA Corporation, Santa Clara, CA, USA), and a Dell P2719H monitor with 1920 × 1080 pixel resolution (Dell Technologies Inc., Round Rock, TX, USA) operating on Windows 10 Pro (Microsoft Corp., Redmond, WA, USA). High-resolution images were used to ensure diagnostic precision.

The criteria used to assess the positioning of impacted third molars in the cone-beam computed tomography images were based on the classifications of Winter [19] and Pell and Gregory [20].

Winter’s classification [19] assesses the angulation of the long axis of the unerupted third molar in relation to the long axis of the adjacent second molar: 1—vertical—the long axis of the third molar follows the same direction as the long axis of the adjacent second molar; 2—horizontal—the long axis of the third molar is perpendicular to the long axis of the second molar; 3—mesioangular—the crown of the third molar is inclined toward the second molar; 4—distoangular—the long axis of the third molar is inclined distally to the second molar; 5—inverted—the crown assumes the opposite direction to the occlusal plane; 6—vestibuloversion—the crown of the third molar faces the buccal surface; 7—linguoversion—the crown of the third molar faces the lingual face.

Pell and Gregory’s classification [20] comprises two criteria. The first criterion is the depth of the tooth in the arch concerning the occlusal plane: 1—Position A—the surface of the third molar is at the level or above the occlusal plane of the adjacent second molar; 2—Position B—the surface of the third molar between the occlusal plane and the cervical line of the adjacent second molar; 3—Position C—the surface of the third molar is located below the cervical line of the adjacent second molar. The second criterion is the depth of the tooth in the ramus of the mandible: 1—Class I—the third molar crown is located completely in front of the anterior edge of the ramus; 2—Class II—the third molar crown is partially inside the mandibular ramus; 3—Class III—the third molar crown is completely within the mandibular ramus.

The findings related to the relationship of the third molar root apex with the mandibular canal (inferior alveolar nerve) were assessed based on the following aspects [21]: 1—the superimposition of the root apex of the third molar with the mandibular canal; 2—the presence of contact between the root apex of the third molar and the mandibular canal (inferior alveolar nerve); 3—absence of contact between the root apex of the third molar and the mandibular canal (inferior alveolar nerve).

The findings related to external root resorption in the distal part of the lower second molar were derived from a prior study that examined a similar condition for upper teeth [17]: 0—the absence of external root resorption; 1—the presence of external root resorption.

All CBCT scans were standardized to ensure axial alignment of the third molar. The sagittal and coronal planes were utilized to orient the long axis of the sample transversely to the ground, correcting for the parallax error. Analysis of the CBCT images employed a specific filter within the e-Vol DX software [16]. The map-reading strategy in CBCT scans was executed in axial, sagittal, and coronal sections with a resolution of 0.1 × 0.1 mm, spanning from the most coronal point of the impacted tooth to the most apical point of the root apex. Dynamic navigation in CBCT scans encompassed all adjacent regions, including the mandibular second molar, the relationship of the apex of the impacted tooth to the mandibular canal, and the relationship between the impacted tooth and the mandibular ramus. The precise positioning of the mandibular third molar was carefully observed and tabulated.

All analyses were conducted collaboratively by two examiners, each possessing over ten years of experience in interpreting cone-beam computed tomography scans. The examiners underwent prior calibration by reviewing exams that adhered to the study’s inclusion and exclusion criteria, representing a total of 10% of the sample. In instances where consensus was lacking, a third examiner, equally qualified, was consulted to make the final decision.

The statistical treatment used described the variables as frequencies and percentages, using the Jamovi software—1.6.23.0 (The Jamovi Project, 2019). Fisher’s exact test was used for a comparative statistical analysis between the position of the apex of the impacted third molar/mandibular canal in both classifications used. The significance level considered for the analysis was set at 5%.

## 3. Results

### 3.1. Patient Characteristics

In the 200 imaging exams assessed, a total of 310 impacted third molars was identified. The distribution of teeth by gender revealed that 55.16% (171) were observed in female individuals, while 44.83% (139) were in males. Regarding age groups, 47.7% (148) were between 18 and 30 years old, 43.9% (136) were between 31 and 50 years old, and 8.4% (26) were between 51 and 80 years old. The distribution of impacted teeth showed a proportion of 51.3% (159) for tooth 38 and 48.7% (151) for tooth 48.

### 3.2. Positioning of Impacted Third Molars

According to Winter’s classification [19], which assesses the alignment of the long axis of the third molar with that of the second molar, the observations revealed that 48.06% of impacted teeth were in the mesioangular position. This was followed by the vertical, distoangular, horizontal, and vestibuloversion positions. Notably, no inverted or linguoversion teeth were observed in this sample (Table 1).

According to Pell and Gregory’s classification [20], the analysis involved two criteria. In examining the depth of the bone where the impacted third molar was located, it was observed that 43.22% of the teeth were in position B (the surface of the third molar between the occlusal plane and the cervical line of the second molar) and 43.22% were in position C (the surface of the third molar below the cervical line of the adjacent second molar) (Table 2). When evaluating the available space between the distal part of the second molar and the ramus of the mandible, 54.2% of the teeth were Class II (the crown of the third molar partially inside the ramus of the mandible) (Table 3) (Figure 1).

### 3.3. Anatomical Relationship between the Root Apex of the Impacted Third Molar and the Mandibular Canal

In the evaluated sample, considering the anatomical position between the root apex of the impacted lower third molar and the mandibular canal, it was observed that in 69.7% of the cases, there was an absence of contact. In 22.9% of the cases, a superposition of the third molar root apex with the mandibular canal was noted, and in 7.4% of cases, there was direct contact between the third molar root apex and the mandibular canal (Table 4) (Figure 2).

Teeth classified as Class II, position B (crown of the third molar partially inside the ramus of the mandible and the surface of the third molar between the occlusal plane and the cervical line of the second molar), according to Pell and Gregory [20], exhibited a higher incidence of overlap with the mandibular canal. Similarly, Class III, position C (those in which the crown of the third molar was completely within the ramus of the mandible and the surface of the third molar was below the cervical line of the adjacent second molar), also showed a notable association with overlap with the mandibular canal. Table 5 presents the relationship between the position of the apex of the impacted third molar and the mandibular canal according to Pell and Gregory’s [20] classification, while Table 6 presents the relationship between the position of the apex of the impacted third molar and the mandibular canal according to Winter’s classification [19].

### 3.4. Occurrence of External Root Resorption on the Distal Side of the Mandibular Second Molar

The analysis revealed external root resorption in the distal part of the lower second molar in 11.3 % of the cases, whereas in 88.7% of the samples, no such occurrence was observed. Among the impacted lower third molars, tooth 48 was identified as the primary contributor to external root resorption in the distal part of the lower second molar, accounting for 7.41%, followed by tooth 38 at 3.87% (Figure 3).

## 4. Discussion

Understanding the spatial position and anatomical characteristics associated with impacted third molars is crucial for diagnosing, planning, and monitoring surgical interventions, leading to a reduction in complications. Accurate surgical planning for impacted teeth helps to prevent various complications during the operative procedures and minimizes potential legal issues. The utilization of a map-reading strategy, particularly with modern cone-beam computed tomography software, enhances the identification of anatomical positions and characteristics of impacted third molars. This not only boosts professional confidence but also sets a standard for the quality and safety of surgical procedures for the benefit of the patient. Evidence indicates that the morbidity and surgical risk associated with the removal of impacted teeth depend on operative factors, including flap design and the utilization of a minimally invasive approach [22,23].

Panoramic radiography has traditionally been employed for investigating third molars, providing a broad overview of teeth and bone structures [8,9,10]. This imaging technique is valued for its simplicity, cost-effectiveness, and comprehensive analysis of the oral and maxillofacial complex in a population. According to Kim et al. [10], the accurate interpretation of panoramic radiography can offer clinicians various signs that assist in predicting the position of wisdom teeth, ultimately reducing the risk of injury to the inferior alveolar nerve. However, conventional radiographic methods have limitations, underscoring the necessity for more advanced exams that deliver clearer, more accurate, and detailed information. Additionally, the ability to differentiate between soft and hard tissues and visualize specific areas for precise sections or cuts is a distinct advantage offered by cone-beam computed tomography [11,12,13,14,15,16,17,18,24].

The relationship between impacted third molars and the mandibular canal in panoramic radiographs and cone-beam computed tomography was previously evaluated in [12], and the results showed that cone-beam computed tomography provides more accurate information. Peker et al. [14] analyzed the correlation between cone-beam computed tomography and digital panoramic radiography in detecting the number of roots of impacted third molars and the relationship of these roots with the mandibular canal. The number of roots could not be accurately determined using panoramic radiographs, and cone-beam computed tomography was necessary in the preoperative evaluation of impacted third molars when the darkening of the roots and interruption of the margins of the mandibular canal were observed on panoramic radiographs. Moreover, studies such as that by Baena et al. [15] emphasize that in cases with potential risks, cone-beam computed tomography enables a superior assessment of the anatomical relationship between the mandibular canal and the third molar root, allowing for the observation of cortical bone around the inferior alveolar nerve.

Panoramic radiography tends to underestimate the space available for the accommodation of the third molar when compared to cone-beam computed tomography [25]. The impact of cone-beam computed tomography examinations on the diagnosis and treatment of mandibular third molars among oral and maxillofacial surgeons is evident in the increased level of professional confidence in clinical judgment. The use of cone-beam computed tomography has significantly improved confidence levels in the diagnosing and treating of these teeth, along with an enhanced perception of the surgical complexity involved. This underscores the importance of considering cone-beam computed tomography in the diagnosis and treatment planning for lower third molars [18].

Several details of the methodology were carefully considered in this study. The minimum age for participants in this study was set at 18 years, taking into account reports from the literature that indicated jaw growth completion by the age of 17 years of age [26]. An essential aspect of the methodology involves the analysis of cone-beam computed tomography exams. Utilizing a model employed in previous studies for navigating through the cone-beam computed tomography images enhances visualization and facilitates the identification of aspects to be analyzed. This method ensures a multidimensional visualization, examining the images millimeter by millimeter and in all planes [27].

In the present study, it was observed that 55.1% of impacted third molars were found in female individuals. The predominant age group was 18 to 30 years old (47.7%), and the distribution of impacted third molars indicated 51.3% for tooth 38 (mandibular left third molar) and 48.7% for tooth 48 (third molar bottom right). These findings are in line with previous studies [1,12,28]. Examining the spatial position, based on Winter’s classification [19], 48.06% of impacted teeth were in the mesioangular position. According to Pell and Gregory’s classification [20], 43.22% of the teeth were in position B (the surface of the third molar between the occlusal plane and the cervical line of the adjacent second molar) and position C (the surface of the third molar located below the cervical line of the adjacent second molar). When assessing the space between the distal part of the second molar and the ramus of the mandible, 54.2% of the teeth were classified as Class II, and 40.32% were classified as Class III [19]. In addition, 69.7% of teeth observed showed an absence of contact between the root apex and the mandibular canal; in 22.9%, there was a superimposition; and in 7.4%, there was contact between the third molar root apex and the mandibular canal. Gu et al. [29] reported that 92.7% of third molars studied had no direct contact with the mandibular canal. Tassoker [30] evaluated the relationship between the mandibular canal and the impacted mandibular third molar using panoramic radiography and cone-beam computed tomography. The analysis showed that when the deviation of the mandibular canal is observed, cone-beam computed tomography exam is recommended, as it allows the reduction of the risk of injury to the inferior alveolar nerve. Recent studies [18,31] also demonstrated the superiority of cone-beam computed tomography in detecting details related to impacted third molars, such as the presence of second molar resorption and contact with the inferior alveolar nerve. Additionally, there is an increased level of confidence in the diagnosis and treatment planning for mandibular third molars with the use of cone-beam computed tomography.

The resorption of the distal surface of the second molar, caused by an impacted third molar, is considered an irreversible pathology. According to the National Institute for Health and Care, when this pathology is observed, the intervention of choice is the surgical removal of the impacted third molar. Most studies comparing the use of imaging exams to investigate the presence of external root resorption in second molars adjacent to impacted third molars reported a low prevalence ranging from 0.3 to 7% [32,33]. However, this study observed a higher incidence, with an index of 11.3% for the occurrence of external root resorption in the distal part of the mandibular second molar. Oenning et al. [5,34] reported a 14.3% prevalence of external root resorption in the distal second molar using cone-beam computed tomography, and Li et al. [35] showed a higher prevalence of 52.9%.

The present study has some limitations, including the collection of cone-beam computed tomography scans from only one imaging center and the retrospective nature of the evaluation. More reliable and comprehensive information can be obtained through data collection from multiple centers and by incorporating longitudinal evaluations.

The significance and clinical application of this study lie in its contribution to improved planning, diagnosis, and prognosis, leading to decision-making with greater confidence for the dental professional and enhanced comfort and safety for the patient. The potential reduction of unpleasant intercurrences, including the risk of accidents such as injury to the neurovascular bundle within the mandibular canal, along with a more predictable analysis of buccal and lingual bone thickness, are critical aspects that should be considered when planning surgeries for mandibular third molars.

## 5. Conclusions

Based on dynamics of navigation, it was observed that 48.06% of impacted teeth were in the mesioangular position; 69.7% showed an absence of contact between the root apex and the mandibular canal; 22.9% exhibited a superimposition; and in 7.4%, there was contact between the third molar root apex and the mandibular canal. The application of navigation dynamics in cone-beam computed tomography images, utilizing sophisticated post-processing software for the analysis of anatomical positions of impacted third molars, serves to enhance professional confidence and establishes a higher standard of quality and safety in the surgical procedure for the patient.

## Figures and Tables

**Figure 1 diagnostics-14-00260-f001:**
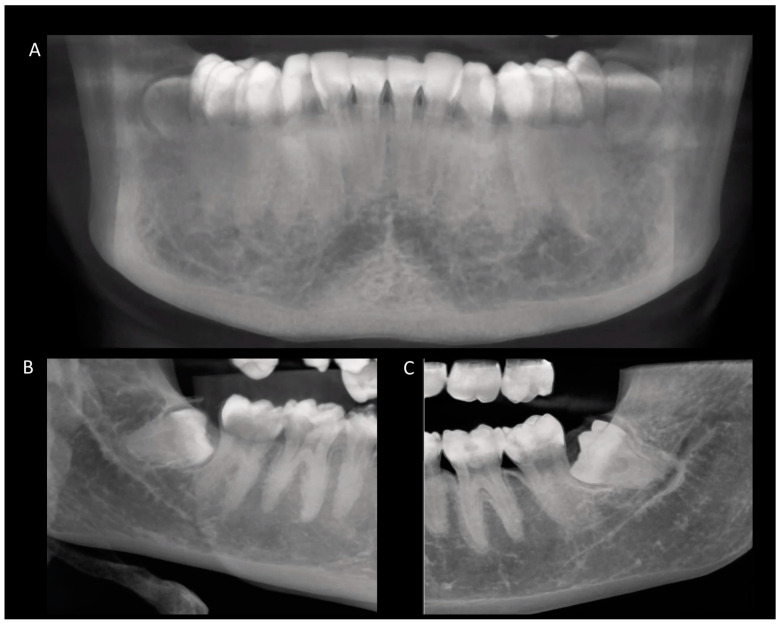
CBCT scans showing an overview of the spatial position and anatomical characteristics related to impacted third molars (**A**–**C**).

**Figure 2 diagnostics-14-00260-f002:**
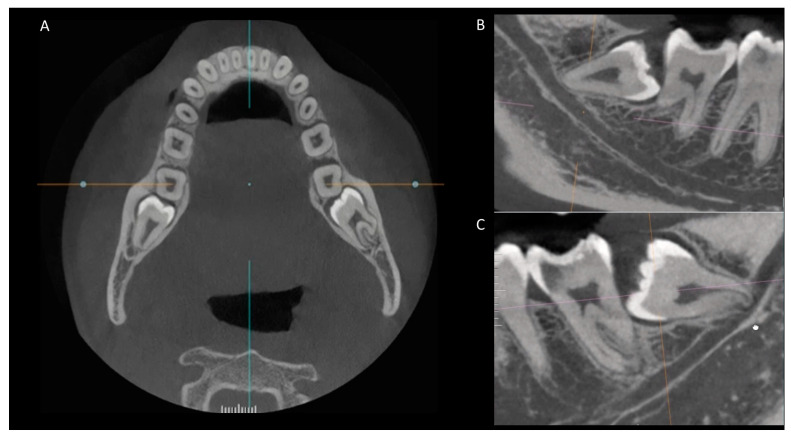
CBCT scans showing the anatomical relationship between the root apex of the impacted third molar and the mandibular canal (**A**–**C**).

**Figure 3 diagnostics-14-00260-f003:**
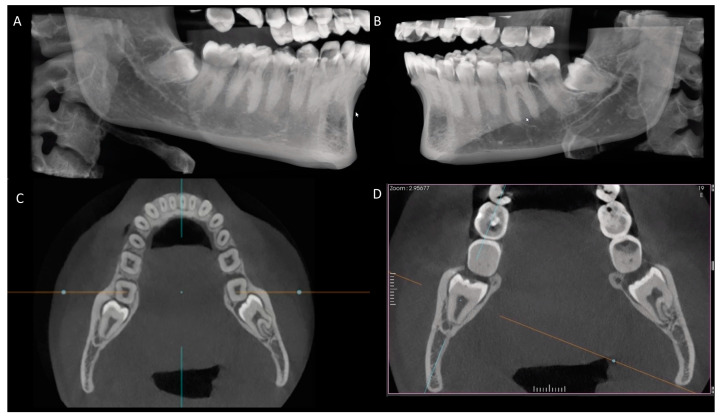
CBCT scans showing the distal side of the mandibular second molar and the possibility of root resorption associated with impacted third molar (**A**–**D**).

**Table 1 diagnostics-14-00260-t001:** Distribution of impacted teeth according to Winter’s classification.

Winter’s Classification	*N*	%
Mesioangular	149	48.06
Vertical	86	27.74
Distoangular	64	20.64
Horizontal	6	1.93
Vestibuloversion	5	1.61
Inverted	0	0
Linguoversion	0	0

**Table 2 diagnostics-14-00260-t002:** Distribution of impacted teeth according to the Pell and Gregory classification—depth in the bone.

Pell and Gregory Classification	*N*	%
Position A	42	13.54
Position B	134	43.22
Position C	134	43.22

**Table 3 diagnostics-14-00260-t003:** Distribution of impacted teeth according to Pell and Gregory’s classification—existing space between the distal part of the second molar and the ramus of the mandible.

Pell and Gregory’s Classification	*N*	%
Class I	17	5.48
Class II	168	54.2
Class III	125	40.32

**Table 4 diagnostics-14-00260-t004:** Anatomical relationship between the root apex of the impacted third molar and the mandibular canal.

Anatomical Relationship	*N*	%
Superposition	71	22.9
Apical contact	23	7.4
No contact	216	69.7

**Table 5 diagnostics-14-00260-t005:** Relationship between the position of the apex of the impacted third molar/mandibular canal in Pell and Gregory’s classification.

Relation	Class	Position			*p*-Value
		A N (%)	B N (%)	C N (%)	
Superposition	Class I	0	1 (0.32)	3 (0.96)	˂0.001
	Class II	4 (1.29)	25 (8;06)	8 (2.58)	
	Class III	0	5 (1.61)	25 (8.06)	
Apical contact	Class I	0	0	0	0.63
	Class II	0	7 (2.25)	3 (0.96)	
	Class III	0	4 (1.29)	9 (2.90)	
No contact	Class I	1 (0.32)	4 (1.29)	8 (2.58)	˂0.001
	Class II	37 (11.93)	67 (21.6)	17 (5.48)	
	Class III	0	21 (6.77)	61 (19.67)	

**Table 6 diagnostics-14-00260-t006:** Relationship between the position of the apex of the impacted third molar/mandibular canal in Winter’s classification.

Relation	Position					
	Vertical	Horizontal	Mesioangular	Distoangular	Vestibuloversion	*p*-Value
Superposition	28	1	33	8	1	0.002
Apical contact	3	4	15	1	0	
No contact	55	1	101	55	4	

## Data Availability

The data are available in this article.

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
