# Peer review of "Spatial Position and Anatomical Characteristics Associated with Impacted Third Molars Using a Map-Reading Strategy on Cone-Beam Computed Tomography Scans: A Retrospective Analysis"

_diagnostics, 2024, doi:10.3390/diagnostics14030260_

Round 1
Reviewer 1 Report
Comments and Suggestions for Authors
Authors evaluated the spatial position and anatomical characteristics related to impacted third molars by map-reading strategy using cone beam computed tomography.
Third molar diagnostics and safe surgery is an importan issue in dento-alveolar surgery.
Many classifications, evalutaing methods were decribed in the last 100 years, to detect position of the wisdoms and/or localize the nerve around the tip of the roots, around the teeth.
The discribed method is simple, cheap, safe, and easy to use in the daily clinical practice.
The article is a tipical example for those papers, wich do not want to use complicated methods, do not perform mutivariant statistics, are not based on expensive tools.
The statements are clear, helps the clinicians in the daily work. The advised method provides safety for the paient wich should be the only goal of a surgeon.
I advise the publicaion without any changes.
Author Response
Reviewer 1 - Comments |
Answers |
Authors evaluated the spatial position and anatomical characteristics related to impacted third molars by map-reading strategy using cone beam computed tomography. Third molar diagnostics and safe surgery is an important issue in dento-alveolar surgery. Many classifications, evaluating methods were described in the last 100 years, to detect position of the wisdoms and/or localize the nerve around the tip of the roots, around the teeth. The described method is simple, cheap, safe, and easy to use in the daily clinical practice. The article is a typical example for those papers, which do not want to use complicated methods, do not perform multivariant statistics, are not based on expensive tools. The statements are clear, helps the clinicians in the daily work. The advised method provides safety for the patient which should be the only goal of a surgeon. I advise the publication without any changes. |
Thanks for the comments and suggestions. |

Reviewer 2 Report
Comments and Suggestions for Authors
Dear author, the article is very interesting, and the topic is very hot in dentistry and oral surgery.
I am honored to give you just few suggestions to improve the article.
The title is fine and well done but I would add also the type of the study in it, for example “Retrospective study”.
The present study is well designed and conducted, and the manuscript is quite clear.
The English is good and spelled correctly but to be published it needs of some adjustment.
There are some comments below.
Abstract: The abstract correctly summarizes the study design and purpose as the title as well.
Keywords: The keywords are correct and perfectly fitting the study design, I would add “inferior third molar” if it is possible.
Introduction:
- Line 50-56: You should improve your references adding also other clinical articles contrasting what you reported about Orthopantomography (OPT) and CBCT. Some South Korean authors collaborating with Kim YS reported, that when there is a translucency in correspondence of the apical portion of the third molar it would be near or impacted in the lingual cortical plate, confirming this radiological sign in CBCT. And when the risk of lingual cortical plate or mandibular canal damage is high, he reported good results and long follow-up of “coronectomy procedures” leaving in place the roots. So you should reported that CBCT is considered the best exam for an anatomical structural evaluation but it is a second level exam after an evaluation of OPT because there are also radiological sign in OPT to evaluate the potential risks of the intervention and the CBCT needs. In this case you should cite these following articles:
Kim, Y.-S.; Park, Y.-M.; Cosola, S.; Riad, A.; Giammarinaro, E.; Covani, U.; Marconcini, S. Retrospective Analysis on Inferior Third Molar Position by Means of Orthopantomography or CBCT: Periapical Band-Like Radiolucent Sign. Appl. Sci. 2021, 11, 6389. https://doi.org/10.3390/app11146389.
Cosola S, Kim YS, Park YM, Giammarinaro E, Covani U. Coronectomy of Mandibular Third Molar: Four Years of Follow-Up of 130 Cases. Medicina (Kaunas). 2020 Nov 27;56(12):654. doi: 10.3390/medicina56120654.
Cervera-Espert J, Pérez-Martínez S, Cervera-Ballester J, Peñarrocha-Oltra D, Peñarrocha-Diago M. Coronectomy of impacted mandibular third molars: A meta-analysis and systematic review of the literature. Med Oral Patol Oral Cir Bucal. 2016 Jul 1;21(4):e505-13. doi: 10.4317/medoral.21074.
Line 72-76: I do not think patients with bisphosphonate intake were included in the study, or yes? Please add this aspect in the exclusion criteria following last review on the extraction in patients with bisphosphonate therapy so you should cite in this case the following systematic review.
Dioguardi M, Di Cosola M, Copelli C, Cantore S, Quarta C, Nitsch G, Sovereto D, Spirito F, Caloro GA, Cazzolla AP, Aiuto R, Cascardi E, Greco Lucchina A, Lo Muzio L, Ballini A, Mastrangelo F. Oral bisphosphonate-induced osteonecrosis complications in patients undergoing tooth extraction: a systematic review and literature updates. Eur Rev Med Pharmacol Sci. 2023 Jul;27(13):6359-6373. doi: 10.26355/eurrev_202307_32996.
Discussion needs just few references to be added, you should mention also authors with different view from yours and you should mention also the fact that the morbility and the surgical risk of the surgery may depend also on the flap design and the minimally invasive approach.
Glera-Suárez P, Soto-Peñaloza D, Peñarrocha-Oltra D, Peñarrocha-Diago M. Patient morbidity after impacted third molar extraction with different flap designs. A systematic review and meta-analysis. Med Oral Patol Oral Cir Bucal. 2020 Mar 1;25(2):e233-e239. doi: 10.4317/medoral.23320.
Gay-Escoda C, Sánchez-Torres A, Borrás-Ferreres J, Valmaseda-Castellón E. Third molar surgical difficulty scales: systematic review and preoperative assessment form. Med Oral Patol Oral Cir Bucal. 2022 Jan 1;27(1):e68-e76. doi: 10.4317/medoral.24951. PMID: 34874928; PMCID: PMC8719785.
Anyway, the rest of discussion and conclusion are well done.
Please explain more about the limitation of this study in the discussion, also about the retrospective nature of the study because admitting the limitation is a good aspect of a clinical study bringing researcher to improve for future study.
I hope these suggestions may help you to publish the article.
Best regard,
Author Response
Reviewer 2 - Comments |
Answers |
Dear author, the article is very interesting, and the topic is very hot in dentistry and oral surgery.
I am honored to give you just few suggestions to improve the article.
|
Thanks for the comments and suggestions. |
The title is fine and well done but I would add also the type of the study in it, for example “Retrospective study”.
|
Ok. The title was modified following the reviewer’s recommendations. |
The present study is well designed and conducted, and the manuscript is quite clear.
The English is good and spelled correctly but to be published it needs of some adjustment.
|
Ok. English has undergone further revision. |
Abstract: The abstract correctly summarizes the study design and purpose as the title as well.
|
Thanks for the comments and suggestions. |
Keywords: The keywords are correct and perfectly fitting the study design, I would add “inferior third molar” if it is possible.
|
Ok. New keyword has been added. |
Introduction: - Line 50-56: You should improve your references adding also other clinical articles contrasting what you reported about Orthopantomography (OPT) and CBCT. Some South Korean authors collaborating with Kim YS reported, that when there is a translucency in correspondence of the apical portion of the third molar it would be near or impacted in the lingual cortical plate, confirming this radiological sign in CBCT. And when the risk of lingual cortical plate or mandibular canal damage is high, he reported good results and long follow-up of “coronectomy procedures” leaving in place the roots. So you should reported that CBCT is considered the best exam for an anatomical structural evaluation but it is a second level exam after an evaluation of OPT because there are also radiological sign in OPT to evaluate the potential risks of the intervention and the CBCT needs. In this case you should cite these following articles:
Kim, Y.-S.; Park, Y.-M.; Cosola, S.; Riad, A.; Giammarinaro, E.; Covani, U.; Marconcini, S. Retrospective Analysis on Inferior Third Molar Position by Means of Orthopantomography or CBCT: Periapical Band-Like Radiolucent Sign. Appl. Sci. 2021, 11, 6389. https://doi.org/10.3390/app11146389.
Cervera-Espert J, Pérez-Martínez S, Cervera-Ballester J, Peñarrocha-Oltra D, Peñarrocha-Diago M. Coronectomy of impacted mandibular third molars: A meta-analysis and systematic review of the literature. Med Oral Patol Oral Cir Bucal. 2016 Jul 1;21(4):e505-13. doi: 10.4317/medoral.21074.
|
Ok. References have been included. |
Line 72-76: I do not think patients with bisphosphonate intake were included in the study, or yes? Please add this aspect in the exclusion criteria following last review on the extraction in patients with bisphosphonate therapy so you should cite in this case the following systematic review.
Dioguardi M, Di Cosola M, Copelli C, Cantore S, Quarta C, Nitsch G, Sovereto D, Spirito F, Caloro GA, Cazzolla AP, Aiuto R, Cascardi E, Greco Lucchina A, Lo Muzio L, Ballini A, Mastrangelo F. Oral bisphosphonate-induced osteonecrosis complications in patients undergoing tooth extraction: a systematic review and literature updates. Eur Rev Med Pharmacol Sci. 2023 Jul;27(13):6359-6373. doi: 10.26355/eurrev_202307_32996.
|
The analysis conducted in the present study was retrospective and relied on the examination of cone-beam computed tomography (CBCT) scans obtained from a private clinic database. Unfortunately, information regarding the patients' medical history was not available. As a result, this type of information was not utilized in the establishment of inclusion and exclusion criteria. |
Discussion needs just few references to be added, you should mention also authors with different view from yours and you should mention also the fact that the morbility and the surgical risk of the surgery may depend also on the flap design and the minimally invasive approach.
Glera-Suárez P, Soto-Peñaloza D, Peñarrocha-Oltra D, Peñarrocha-Diago M. Patient morbidity after impacted third molar extraction with different flap designs. A systematic review and meta-analysis. Med Oral Patol Oral Cir Bucal. 2020 Mar 1;25(2):e233-e239. doi: 10.4317/medoral.23320.
Gay-Escoda C, Sánchez-Torres A, Borrás-Ferreres J, Valmaseda-Castellón E. Third molar surgical difficulty scales: systematic review and preoperative assessment form. Med Oral Patol Oral Cir Bucal. 2022 Jan 1;27(1):e68-e76. doi: 10.4317/medoral.24951. PMID: 34874928; PMCID: PMC8719785.
|
Ok. The requested information has been entered.
|
Anyway, the rest of discussion and conclusion are well done.
|
Thanks for the comments and suggestions. |
Please explain more about the limitation of this study in the discussion, also about the retrospective nature of the study because admitting the limitation is a good aspect of a clinical study bringing researcher to improve for future study.
|
Ok. Information about the limitations of the study was included in the discussion. |
